# Organisational and individual readiness for change to respectful maternity care practice and associated factors in Ibadan, Nigeria: a cross-sectional survey

Oluwaseun Taiwo Esan  ,[1,2] Salome Maswime,[3] Duane Blaauw[2]

¹Department of Community Health, Faculty of Clinical Sciences, Obafemi Awolowo University, Ile-Ife, Osun State, Nigeria
²Centre for Health Policy, School of Public Health, Faculty of Health Sciences, University of the Witwatersrand, Johannesburg, Gauteng, South Africa
³Global Surgery Division, Department of Surgery, Faculty of Health Sciences, University of Cape Town, Cape Town, Western Cape, South Africa

**Correspondence to**
Dr Oluwaseun Taiwo Esan;
o.esan@oauife.edu.ng

## ABSTRACT

**Objectives** This study assessed health providers' organisational and individual readiness for change to respectful maternity care (RMC) practice and their associated factors in Ibadan Metropolis, Nigeria.

**Design** A cross-sectional survey using standardised structured instruments adapted from the literature.

**Setting** Nine public health facilities in Ibadan Metropolis, Nigeria, 1 December 2019–31 May 2020.

**Participants** 212 health providers selected via a two-stage cluster sampling.

**Outcomes** Organisational readiness for change to RMC ($ORC_{RMC}$) and individual readiness for change to RMC ($IRC_{RMC}$) scales had a maximum score of 5. Multiple linear regression was used to identify factors influencing $IRC_{RMC}$ and $ORC_{RMC}$. We evaluated previously identified predictors of readiness for change (change valence, informational assessments on resource adequacy, core self-evaluation and job satisfaction) and proposed others (workplace characteristics, awareness of mistreatment during childbirth, perceptions of women's rights and resource availability to implement RMC). Data were adjusted for clustering and analysed using Stata V.15.

**Results** The providers' mean age was 44.0±9.9 years with 15.4±9.9 years of work experience. They scored high on awareness of women's mistreatment (3.9±0.5) and women's perceived rights during childbirth (3.9±0.5). They had high $ORC_{RMC}$ (4.1±0.9) and $IRC_{RMC}$ (4.2±0.6), both weakly but positively correlated (r=0.407, 95% CI: 0.288 to 0.514, p<0.001). Providers also had high change valence (4.5±0.8) but lower perceptions of resource availability (2.7±0.7) and adequacy for implementation (3.3±0.7). Higher provider change valence and informational assessments were associated with significantly increased $IRC_{RMC}$ (β=0.40, 95% CI: 0.11 to 0.70, p=0.015 and β=0.07, 95% CI: 0.01 to 0.13, p=0.032, respectively), and also with significantly increased $ORC_{RMC}$ (β=0.47, 95% CI: 0.21 to 0.74, p=0.004 and β=0.43, 95% CI: 0.22 to 0.63, p=0.002, respectively). Longer years of work experience (β=0.08, 95% CI: 0.01 to 0.2, p=0.024), providers' monthly income (β=0.08, 95% CI: 0.02 to 0.15, p=0.021) and the health facility of practice were associated with significantly increased $ORC_{RMC}$.

**Conclusion** The health providers studied valued a change to RMC and believed that both they and their facilities were ready for the change to RMC practice.

### STRENGTHS AND LIMITATIONS OF THIS STUDY

⇒ The study was conducted in the pre-implementation phase before the integration of respectful maternity care practice into routine childbirth care in the study location.

⇒ Organisational and individual readiness for change theories were tested quantitatively using very brief standardised assessment scales (12 items and 6 items) among health providers, with zero non-response rate recorded.

⇒ All categories of maternal healthcare providers were interviewed, which may facilitate stakeholder engagement during the implementation process.

⇒ The study was limited in its geographical scope as it was conducted in Ibadan Metropolis, one metropolitan area in a Southwestern Nigerian state.

⇒ The study was further limited in its scope as tertiary health facilities were not studied because there was only one tertiary health facility serving the populations in the study location.

## INTRODUCTION

The utilisation of maternal care services, especially during childbirth, is low in Nigeria. The proportion of women whose delivery used a skilled birth attendant in 2018 was 43.3%.[1] One of the reasons explaining this is women's mistreatment during birth.[2 3] Negative health worker attitudes have been expressed as mistreatment, particularly during childbirth, and this has been reported frequently, both globally and in Nigeria specifically. Ogunlaja *et al*[4] found a 93.2% reported prevalence of mistreatment in previous deliveries among 438 antenatal clients in Ogbomoso, Oyo state. The prevalence of Nigerian women's mistreatment during childbirth was reported as ranging from 11% to 71% according to a systematic review of 14 studies between 2004 and 2015.[5] Respectful maternity care (RMC) practices have been prioritised as a means to improve patient–provider interactions and the quality of maternal care experienced.

RMC has been defined as 'care organised for and provided to all women in a manner that maintains their dignity, privacy and confidentiality, ensures freedom from harm and mistreatment, and enables informed choice and continuous support during childbirth' (page 3).[6] It is a human rights approach to maternity care[7] and is recommended as the standard for all women.[8] Several RMC-promoting interventions have been implemented and have shown promising results.[9] For these results to be enduring and sustainable, the health providers will need to embrace and support the interventions. This can be achieved if they are ready for the change to an RMC practice.

Readiness for change measures the extent to which people or organisations are inclined to adopt a change that alters the 'status quo'.[10] It addresses the psychological and behavioural forms of readiness for change, that is, the state of being willing and able to change.[11 12] Some authors also describe it as having a structural component that addresses the presence or absence of financial, material and human resources needed for a change, such as to RMC practice.[12] Readiness for change is a multilevel construct measured at individual and organisational levels. Organisational readiness for change is a multifaceted concept that consists of employees' change commitment (collective resolve) and change efficacy (perceived shared ability) to implement the change.[11] Individual readiness for change is an employee's confidence to manage the change or willingness to accept new roles and adopt new practices.[13] Readiness for change is different from preparedness as the latter addresses the set activities to implement the change,[14] while readiness measures being both prepared and motivated to implement the change.

Readiness for change is a key determinant of implementation success.[15 16] The readiness for change theories have been applied in both health and non-health organisations; however, there are no previous studies on their application to RMC-promoting interventions. Readiness for change to RMC among community, facility and policy stakeholders was mentioned as being responsible for the positive results of an RMC project in Kenya.[17] However, readiness for change was not measured directly in that study.[17] Many RMC-promoting interventions have been conducted without prior assessment of the individual employee or organisational readiness for change.[18 19] If readiness is assessed and found necessary, efforts can be directed at improving it. If otherwise, this suggests the providers' willingness to accept the change irrespective of the work task demands brought by it.

The proposed theory of change for this study is that a high organisational and individual readiness for change would lead to the adoption and institutionalisation of RMC practice which should result in long-term outcomes such as increased health facility delivery. Adoption is the temporary altering of attitudes and behaviours to meet the change expectations. Institutionalisation occurs when the change becomes part of the organisational processes.[10] This is assuming all limiting barriers and contextual factors have been identified and addressed. The barriers and contextual factors have been explored, but the data are yet to be published. This study assessed health providers' organisational and individual readiness for change to RMC practice and their associated factors in Ibadan Metropolis, Nigeria.

## METHODS
### Design, setting and participants
This was a cross-sectional survey conducted from 1 December 2019 to 31 May 2020, in Ibadan Metropolis, Oyo state, Nigeria. Ibadan (the third largest city in Nigeria and the seventh in Africa) was selected, being a more cosmopolitan city. This study was conducted among public healthcare providers from the five Local Government Areas (LGAs) in Ibadan Metropolis. There were 6 public secondary and 26 functional public primary health facilities in the five LGAs with a minimum of 12 deliveries per year at the time of conducting the study. Maternity care services, including delivery services, are offered in all facilities, with more specialised care at secondary health facilities. Doctors and nurses attend deliveries at both primary and secondary health facilities, while community health officers, community health extension workers and health auxiliaries (HAs) only attend deliveries at primary health facilities in the study state.

A two-stage cluster sampling technique was used to select the health facilities and providers in the study LGAs. One primary and one secondary health facility were selected in each LGA using simple random sampling, except in one LGA without a secondary health facility. This gave a total of nine health facilities studied (five primary and four secondary health facilities). There were a total of 244 health providers (as the study population or sampling frame) who could attend deliveries in the study facilities (176 in the 4 secondary facilities and 68 in the 5 primary facilities).

A sample size of 210 health providers was calculated using the one-sample mean test[20] in Stata. This represented 86% of the study population. The parameters used were a change commitment mean of 3.64±0.61 SD, based on a similar study in Switzerland, as a proxy for organisational readiness.[21] The required precision was ±5% about the reference mean, with 90% power and a design effect of 2[22] for the cluster sampling. The number of health providers interviewed at each facility was allocated proportionately to the generated total number of health providers per professional type at each health facility within the LGAs.

All the available and consenting health providers at each health facility were interviewed until the required numbers of each professional type for each facility were reached. As the health workers work in shifts (and thus may not have been working at the time of initial approach), if the number to be interviewed was yet to be reached after interviewing all the available and consenting

health workers on the morning and afternoon shifts, the data collectors repeatedly visited the facilities to recruit workers on shifts on later dates. We did not document the number that did not consent, but the majority of those who were approached consented and were interviewed.

## Data collection

Data collection was done using a 112-item tool with 9 sections developed in REDCap and conducted within the health facility premises.[23] Two trained research assistants administered the questionnaire. The tools were pretested among 12 health providers from one public secondary health facility in Ibadan Northwest LGA and one public primary health facility in Ibadan North LGA, after 2-day training. Findings from the pretest were used to improve the data collection instruments. The first part of the instrument assessed health providers' perceptions of women's rights during childbirth, their awareness of women's mistreatment during childbirth in their health facilities and their awareness of the RMC concept. A one-page brief on 'RMC during childbirth' was read to each respondent (see online supplemental file 1). The subsequent sections of the questionnaire evaluated providers' perceptions of individual and organisational readiness for change to RMC practice during childbirth, and possible associated factors, using standardised tools (see online supplemental file 2 for the survey instrument).

The respondents' perceived organisational readiness and individual readiness for change to RMC practice were the outcome variables. Organisational readiness for change to RMC ($ORC_{RMC}$) was assessed using a standardised 12-item Organisational Readiness for Implementing Change (ORIC) tool[24] with five items measuring their change commitment and seven items assessing their change efficacy, both on a 5-point Likert agreement scale.[24] The questions assessing organisational readiness were framed as 'The health workers in this health facility are…'. Organisational readiness was determined as the mean score of the 12 items on the scale with a maximum score of 5. Individual readiness for change to RMC ($IRC_{RMC}$) was measured using a six-item tool on a 5-point Likert agreement scale by Vakola.[13] Questions were framed as 'I am willing to…'. $IRC_{RMC}$ was determined as the mean score of the six-item scale, also with a maximum score of 5. When reported as percentages, the mean scores were standardised and converted using the formula (mean−1)/4×100.

For the predictors, we included factors well described in the implementation science literature and used standardised tools. The previously defined predictors of $ORC_{RMC}$ by Weiner[11] include employee change valence (how much they value the change) and informational assessments (perceived adequacy of the resources available to implement the change). Previously defined predictors of $IRC_{RMC}$ by Vakola[13] include employee job satisfaction and core self-evaluation (which assesses their self-esteem, locus of control, emotional stability

and generalised self-efficacy).[25] We evaluated all of these predictors on both $IRC_{RMC}$ and $ORC_{RMC}$.

In addition, we also proposed that individual provider characteristics such as being younger, having more years of experience and having higher monthly income could positively influence $IRC_{RMC}$ and $ORC_{RMC}$. We suggested that health providers' perceptions about women's rights during childbirth, their perceived availability and adequacy of resources for RMC implementation and differences in their workplace contexts might influence both $IRC_{RMC}$ and $ORC_{RMC}$. Online supplemental file 3 summarises the study's analytical framework and online supplemental file 4 gives the list of standardised tools used to assess the analytical constructs, together with their reliability statistics in our study. The highest Cronbach's α was 0.949 for the organisational readiness for change tool, while the lowest was 0.575 for the tool assessing providers' perception of women's rights.

## Data analysis

Data collected were uploaded to the University of the Witwatersrand data management system via REDCap. Only the first author had access to download and save data on a password-protected computer, then shared them with the coauthors. The dataset has been shared with the Figshare repository.[26]

Data analysis was done using the Stata V.15 software. We adjusted for weighting and facility-level clustering in all analyses using the Stata 'svy' commands. The mean scores of the outcome and predictor variables were determined. Higher mean scores indicate higher $IRC_{RMC}$ and $ORC_{RMC}$. Pearson's correlation was used to evaluate the relationship between $IRC_{RMC}$ and $ORC_{RMC}$, change efficacy and change commitment, and resource availability and adequacy.

Principal component analysis (PCA) was used to construct separate composite indices for the study-specific tools assessing providers' perceptions of women's rights, their awareness of mistreatment in their facilities and the availability of resources for RMC practice. Details are provided in online supplemental file 5. The first components explained 17.9%, 23.2% and 16.5% of the variance for each of these scales, respectively. These PCA scores were then used in the bivariate and multiple regression analyses as potential predictors.

Simple linear regression was done to assess the bivariate relationship between the two numerical outcomes and the predictor variables. Predictors with a p value of ≤0.2 were included in the final multiple regression models for each outcome variable. All predictors were added simultaneously. Multicollinearity analysis was conducted after the regressions. Predictor variables with a high variance inflation factor (>10.0) were excluded from the model.

## Patient and public involvement

A prior qualitative study of pregnant women's perceptions of RMC[27] informed this study, the study location and many of the variables assessed. The women described

**Table 1** Providers' sociodemographic profile by provider type

| Variables | Doctor n=37 | Nurse n=128 | CHEW/CHO n=29 | Auxiliary n=18 | Total n=212 |
|---|---|---|---|---|---|
| Age | | | | | |
| Mean±SD | 38.9±9.9 | 44.6±9.4 | 44.5±9.7 | 49.3±10.5 | 44.0±9.9 |
| Median (IQR) | 40 (31–46) | 44 (39–52) | 46 (39–50) | 52 (40–56) | 44 (38–52) |
| Sex | | | | | |
| Male | 20 (52.3) | 0 (0.0) | 2 (8.5) | 0 (0.0) | 22 (10.4) |
| Female | 18 (47.7) | 127 (100.0) | 26 (91.5) | 18 (100.0) | 190 (89.6) |
| Type of health facility | | | | | |
| Primary | 3 (8.7) | 9 (7.1) | 29 (100.0) | 18 (100.0) | 59 (27.9) |
| Secondary | 34 (91.3) | 119 (92.9) | 0 (0.00 | 0 (0.0) | 153 (72.1) |
| LGA | | | | | |
| Ibadan North | 23 (61.9) | 61 (47.8) | 3 (12.0) | 2 (9.7) | 89 (42.2) |
| Ibadan Northeast | 5 (14.6) | 14 (11.2) | 5 (16.2) | 2 (8.8) | 26 12.3) |
| Ibadan Northwest | 6 (15.6) | 20 (15.9) | 12 (41.6) | 3 (15.0) | 41 (19.3) |
| Ibadan Southeast | 1 (2.9) | 1 (0.8) | 4 (14.8) | 7 (41.8) | 14 (6.6) |
| Ibadan Southwest | 2 (5.1) | 31 (24.3) | 4 (15.3) | 4 (24.7) | 42 (19.7) |
| Study health facility | | | | | |
| Facility 1 | 1 (2.9) | 2 (1.4) | 3 (12.0) | 2 (9.7) | 8 (8.7) |
| Facility 2 | 23 (59.6) | 59 (46.4) | 0 (0.0) | 0 (0.0) | 82 (38.5) |
| Facility 3 | 0 (0.0) | 2 (1.2) | 5 (16.2) | 2 (8.8) | 9 (3.7) |
| Facility 4 | 5 (44.6) | 13 (10.0) | 0 (0.0) | 0 (0.0) | 18 (8.6) |
| Facility 5 | 1 (3.6) | 1 (1.1) | 12 (41.6) | 3 (15.0) | 17 (8.2) |
| Facility 6 | 5 (12.0) | 19 (14.8) | 0 (0.0) | 0 (0.0) | 23 (11.1) |
| Facility 7 | 0 (0.0) | 3 (2.6) | 5 (15.3) | 4 (24.7) | 12 (5.7) |
| Facility 8 | 2 (5.1) | 28 (21.7) | 0 (0.0) | 0 (0.0) | 30 (13.9) |
| Facility 9 | 1 (2.3) | 1 (1.0) | 4 (14.8) | 7 (41.8) | 13 (6.6) |
| Years of experience | | | | | |
| Mean±SD | 10.4±7.7 | 18.2±9.9 | 10.5±7.5 | 13.6±11.0 | 15.4±9.9 |
| Median (IQR) | 10 (3–14) | 18 (11–25) | 8 (4–17) | 9 (6–190) | 14 (7–23) |
| Years working in study facility | | | | | |
| Mean±SD | 3.2±3.5 | 8.2±6.0 | 2.7±1.9 | 3.7±2.0 | 6.0±5.6 |
| Median (IQR) | 2 (0.5–5) | 7 (4–11) | 3 (1–4) | 4 (3–5) | 5 (2–10) |
| Income (in US$) | | | | | |
| Median (IQR) | 658 (526–921) | 500 (289–553) | 270 (132–395) | 99 (26–191) | 463 (263–605) |

CHEW, community health extension worker; CHO, community health officer; IQR, Inter-quartile range; LGA, Local Government Area; SD, Standard deviation.

their experience of childbirth care and queried the readiness of the health providers to provide such care.

## RESULTS
### Sociodemographic profile
Two hundred twelve health providers finally completed the survey, slightly above the required sample size of 210 (with the slight oversampling due to separate data collection by two data collectors). The breakdown by their professional group is shown in table 1. Their overall mean age was 44.0 years. The doctors were the youngest with a mean age (in years) of 38.9, while the HAs were the oldest with a mean age of 49.3. Overall, the respondents had an average of >15 years of post-training work experience, which included an average of about 6 years working at the study facility.

### RMC: women's rights and mistreatment and needed resources
Overall, 35.9% of the providers had heard of RMC. This consisted mainly of doctors (60%) and the least (19.1%) being the HAs. Nonetheless, after RMC had been explained to them, 70% of all the providers agreed that RMC could be implemented in their facilities.

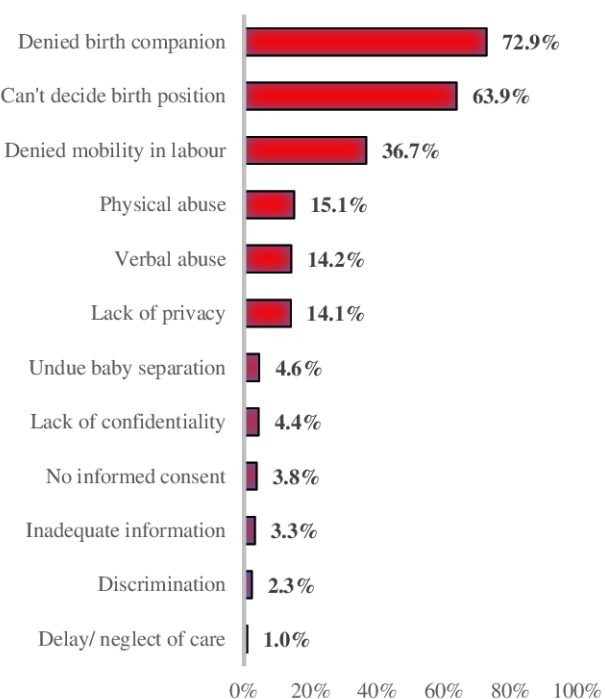

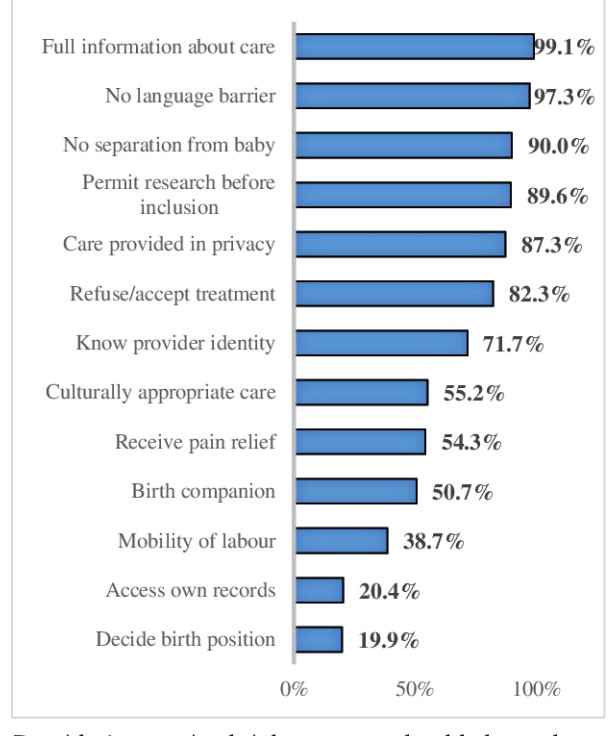

Forms of mistreatment noted by providers in their health facilities (n=212)

Provider's perceived rights women should always have during childbirth (n=212)

**Figure 1** Forms of mistreatment and rights of women during childbirth perceived by providers.

As shown in figure 1, 72.9% of the health providers stated that women delivering in their facility were always denied a birth companion, 63.9% were aware of women not being allowed to decide their birth position and 36.7% had witnessed restrictions on mobility during labour. Correspondingly, only 19.9% of health providers believed that women should always have the right to decide their birth position, 38.7% agreed that women could be mobile during labour and 50.7% supported women having a birth companion (figure 1). Only 20.4% accepted that women should always have unrestricted access to their hospital records.

Figure 2 indicates providers' perceptions of the availability of essential 18 WHO-recommended resources for implementing RMC. The least available resource was RMC educational materials (7.7%), followed by guidelines (8.2%). Approximately 10%–15% of the providers agreed to the availability of private spaces to support birth companions, in-service training on RMC, suggestion boxes and adequately trained staff on RMC. However, 63.0% of them agreed to have curtains and screens for privacy during childbirth.

The mean scores for all the study scales are shown in table 2. The health providers were well aware of the mistreatment of women during childbirth in their health facilities across the 12 items with a high mean score of 3.9±0.5 out of a maximum of 5. However, the mean score of 3.9±0.5 out of 5 also indicates high acceptance of the rights they believe women should always be granted during childbirth.

### Individual and organisational readiness for change to RMC practice

In assessing $ORC_{RMC}$, the health providers scored high on their commitment to the change and their change efficacy, which is their perceived ability to implement the change (table 2). These two constructs were strongly positively correlated (r=0.830, 95% CI: 0.783 to 0.868, p<0.001). Combined, this gave a high mean $ORC_{RMC}$ score of 4.01±0.9, which is 75.3% of the maximum obtainable mean score of 5. The health providers had even higher $IRC_{RMC}$, with a mean score of 4.23±0.6, 80.8% of the maximum. Organisational readiness was only moderately but significantly correlated with $IRC_{RMC}$ (r=0.407, 95% CI: 0.29 to 0.51, p<0.001).

### Change valence and informational assessments

The health providers scored high on how much they value the change to RMC, with a mean of 4.46±0.8 out of 5 (table 2). They, however, scored lower in their informational assessments (3.30±0.7), which describe their perceptions on the adequacy of the available resources to implement the change to RMC practice in their facilities. The providers' mean score for the availability of the WHO-recommended resources to implement RMC was even lower (2.70±0.6), 42.5% of the maximum. There was a mild but significant positive relationship between their perceived availability and adequacy of the resources needed to implement RMC in their facilities (r=0.263, 95% CI: 0.133 to 0.384, p=0.0001). Notwithstanding these perceived deficiencies, the health providers indicated

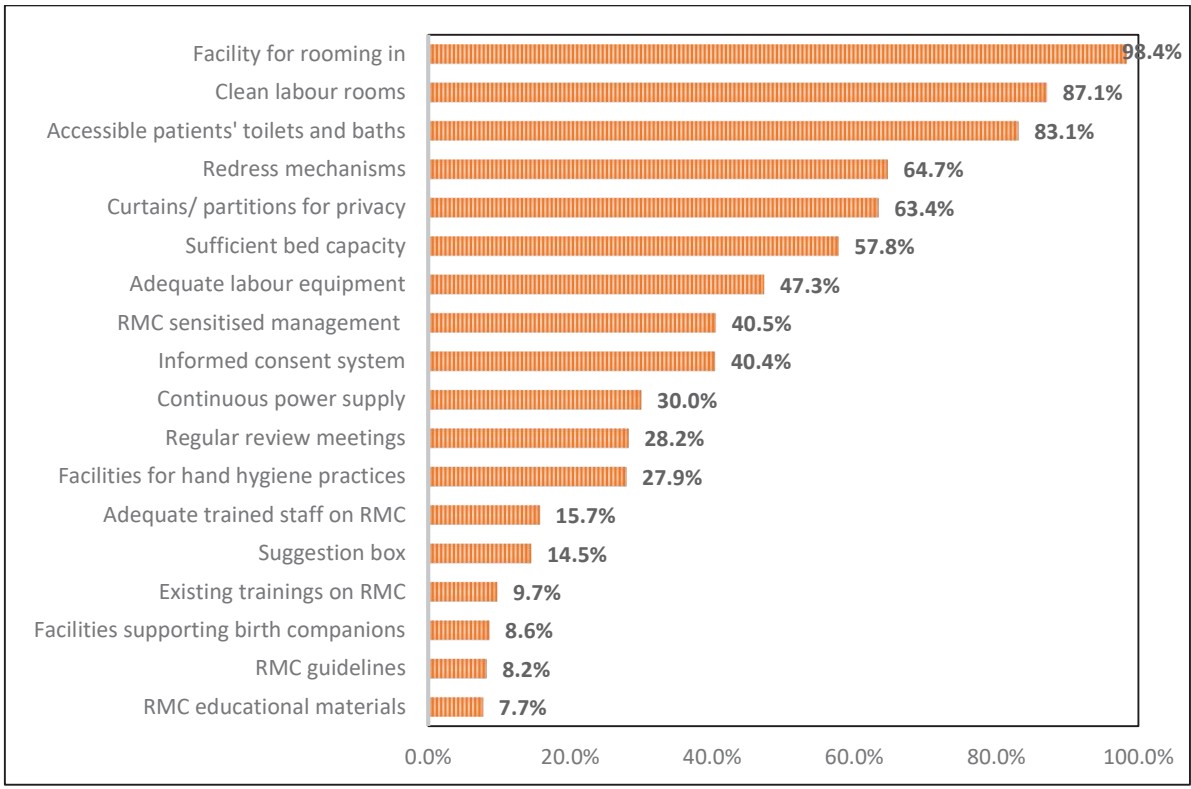

**Figure 2** Provider perceptions on availability of WHO-recommended resources for RMC implementation. RMC, respectful maternity care.

relatively high levels of job satisfaction and core self-evaluation, that is, they had high self-esteem, locus of control, emotional stability and generalised self-efficacy (table 2).

### Factors associated with IRC$_{RMC}$ and ORC$_{RMC}$ practice

Table 3 shows the bivariate and multiple regression analyses for IRC$_{RMC}$, while table 4 shows the analysis for ORC$_{RMC}$. The health providers' change valence and informational assessments were significantly associated with IRC$_{RMC}$ in the multiple regression analysis, increasing

IRC$_{RMC}$ scores (β=0.40, 95% CI: 0.11 to 0.70, p=0.015 and β=0.07, 95% CI: 0.01 to 0.13, p=0.032, respectively). Doctors and nurses had significantly higher IRC$_{RMC}$ than health assistants, in the bivariate analysis, but this was no longer significant after adjusting for other covariates.

IRC$_{RMC}$ varied significantly between health providers from different health facilities in the bivariate analysis but this was no longer the case in the multiple regression analysis. None of the known predictors of individual readiness for change (providers' job satisfaction and core

**Table 2** Average provider perceptions for different study scales (n=212)

| Analytical category | Scale | Mean±SD | 95% CI |
|---|---|---|---|
| Outcomes | Change commitment | 4.05±1.0 | 3.8 to 4.3 |
| | Change efficacy | 3.96±0.9 | 3.6 to 4.3 |
| | Organisational readiness for change | 4.01±0.9 | 3.7 to 4.3 |
| | Individual readiness for change | 4.23±0.6 | 4.1 to 4.4 |
| Predictors | Awareness of mistreatment during childbirth in their facilities | 3.90±0.5 | 3.7 to 4.1 |
| | Women's rights during childbirth | 3.85±0.5 | 3.8 to 4.0 |
| | Change valence | 4.46±0.8 | 4.3 to 4.6 |
| | Informational assessments | 3.30±0.7 | 3.1 to 3.4 |
| | Availability of resources to implement RMC in their facilities | 2.70±0.6 | 2.5 to 2.9 |
| | Core self-evaluation | 4.34±0.5 | 4.3 to 4.4 |
| | Job satisfaction | 3.70±0.6 | 3.6 to 3.8 |

RMC, respectful maternity care.

**Table 3** Analysis of factors associated with health providers' IRC<sub>RMC</sub>

| Covariates | Simple linear regression | | | Multiple linear regression | | |
|---|---|---|---|---|---|---|
| | Crude coefficient | 95% CI | P value | Adjusted coefficient | 95% CI | P value |
| Health providers' age | −0.003 | −0.01 to 0.007 | 0.497 | | | |
| Sex | | | | | | |
| Female | Ref | – | – | Ref | – | – |
| Male | −0.21 | −0.02 to 0.44 | 0.065 | −0.14 | −0.84 to 1.13 | 0.743 |
| Study Local Government Area | | | | | | |
| Ibadan North | Ref | – | – | | | |
| Ibadan Northeast | 0.02 | −0.54 to 0.58 | 0.946 | | | |
| Ibadan Northwest | −0.22 | −0.47 to 0.03 | 0.078 | | | |
| Ibadan Southeast | −0.45 | −052 to 0.37 | **<0.001** | | | |
| Ibadan Southwest | −0.09 | −0.32 to 0.14 | 0.383 | | | |
| Health facility | | | | | | |
| Facility 1 | 0.19 | 0.10 to 0.19 | <0.001 | 0.35 | −0.01 to 0.70 | 0.053 |
| Facility 2 | Ref | – | – | Ref | – | – |
| Facility 3 | −0.53 | −0.53 to −0.53 | <0.001 | −0.06 | −0.39 to 0.27 | 0.675 |
| Facility 4 | 0.29 | 0.29 to 0.29 | <0.001 | 0.07 | −0.05 to 0.18 | 0.218 |
| Facility 5 | −0.37 | −0.37 to −0.37 | <0.001 | −0.23 | −0.51 to 0.04 | 0.087 |
| Facility 6 | −0.06 | −0.06 to −0.06 | <0.001 | 0.04 | −0.07 to 0.14 | 0.423 |
| Facility 7 | 0.12 | 0.12 to 0.12 | <0.001 | 0.22 | −0.13 to 0.54 | 0.175 |
| Facility 8 | −0.10 | −0.10 to −0.10 | <0.001 | 0.0002 | −0.09 to 0.09 | 0.970 |
| Facility 9 | −0.42 | −0.42 to −0.42 | <0.001 | 0.03 | −0.30 to 0.35 | 0.844 |
| Providers' type of health facility | | | | | | |
| Primary | Ref | – | – | | | |
| Secondary | 0.23 | 0.56 to 0.09 | 0.135 | | | |
| Professional cadre | | | | | | |
| Doctor | 0.43 | 0.04 to 0.83 | **0.036** | −0.13 | −0.46 to 0.19 | 0.360 |
| Nurse | 0.37 | 0.05 to 0.70 | **0.030** | Ref | – | – |
| CHEW/CHO | 0.08 | −0.06 to 0.21 | 0.233 | −0.19 | −0.63 to 0.24 | 0.329 |
| Health assistant/aide | Ref | – | – | −0.16 | −0.50 to 0.18 | 0.296 |
| Monthly income (in US$/1000) | −2.36 | −0.06 to 5.28 | 0.097 | 0.05 | −0.03 to 0.13 | 0.187 |
| Years of professional experience | 0.01 | 0.004 to 0.03 | 0.106 | 0.004 | −0.02 to 0.03 | 0.644 |
| Years of experience in the health facility | 0.004 | −0.02 to 0.031 | 0.727 | | | |
| Awareness of the mistreatment of women | 0.01 | −0.04 to 0.05 | 0.712 | | | |
| Perceived women's rights during childbirth | 0.04 | −0.05 to 0.11 | 0.357 | | | |
| Ever heard of RMC (n=170) | | | | | | |
| Yes | −0.02 | −0.33 to 0.30 | 0.883 | | | |
| No | Ref | – | – | | | |
| Perception of RMC being implementable | | | | | | |
| Agreed | 0.10 | −0.34 to 0.53 | 0.62 | | | |
| Indifferent | Ref | – | – | | | |
| Disagreed | 0.06 | −0.46 to 0.58 | 0.794 | | | |
| Change valence (value for RMC practice) | 0.45 | 0.19 to 0.71 | **0.005** | 0.40 | 0.11 to 0.70 | **0.015** |
| RMC informational assessment | 0.07 | 0.15 to 0.42 | **0.001** | 0.07 | 0.01 to 0.13 | **0.032** |

Continued

**Table 3** Continued

| Covariates | Simple linear regression | | | Multiple linear regression | | |
| --- | --- | --- | --- | --- | --- | --- |
| | Crude coefficient | 95% CI | P value | Adjusted coefficient | 95% CI | P value |
| Provider perceptions on available resources | 0.03 | −0.02 to 0.09 | 0.182 | | | |
| Provider job satisfaction | 0.010 | −0.03 to 0.22 | 0.105 | 0.004 | −0.13 to 0.14 | 0.953 |
| Provider core self-evaluation | 0.25 | 0.01 to 0.50 | 0.055 | 0.09 | −0.22 to 0.39 | 0.513 |
| Male # doctor* | | | | 0.15 | −1.01 to 1.31 | 0.765 |
| Constant | | | | 1.76 | 1.37 to 2.14 | **<0.001** |
| | | | | n=212; $R^2$=0.4363; p<0.001 | | |

Predictors with p≤0.2 from the simple linear regression analysis were included in the multiple regression model. The mean variance inflation factor for the multiple regression model is=2.33. Significant p values in bold.

*Male # doctor—interaction between gender and profession.

CHEW, community health extension worker; CHO, community health officer; $IRC_{RMC}$, individual readiness for change to RMC; Ref, reference category; RMC, respectful maternity care.

self-evaluation), nor the newly proposed ones (perceived rights of women, years of experience, income), was significantly associated with $IRC_{RMC}$.

Change valence and informational assessments were also significantly associated with $ORC_{RMC}$ (table 4). A unit increase in the health providers' change valence and informational assessments increased their perceived $ORC_{RMC}$ (β=0.47, 95% CI: 0.21 to 0.74, p=0.004 and β=0.43, 95% CI: 0.22 to 0.63, p=0.002 units, respectively), after adjusting for other covariates. Also, each additional 10 years of work experience significantly increased $ORC_{RMC}$ (β=0.08, 95% CI: 0.01 to 0.2, p=0.024) and each $1000 increase in providers' monthly income increased their perceived $ORC_{RMC}$ (β=0.08, 95% CI: 0.02 to 0.15, p=0.021). There were significant varied associations (positively or negatively) between the health providers' facility of practice and their $ORC_{RMC}$ in relation to the reference facility. The only exception was for facility 4, a secondary health facility in one of the LGAs.

## DISCUSSION

This is the first study to explore individual and organisational readiness for change to RMC practice, and the associated predictors. The health providers had a high level of awareness of the mistreatment of women but also a high general acceptance of women's rights during childbirth. However, there were some rights, such as being allowed a birth companion, that only a few providers regarded as essential, and these were then seldom practised. Nonetheless, the health providers scored high in their perceived $IRC_{RMC}$ and $ORC_{RMC}$. $IRC_{RMC}$ and $ORC_{RMC}$ were only moderately correlated in this analysis. Higher change valence and informational assessment of the adequacy of resources increased not only ORC, as has been found previously,[24 28] but also $IRC_{RMC}$. Job satisfaction and the providers' core self-evaluation, which have been shown to influence IRC,[13 29] had no statistically significant effect on $IRC_{RMC}$ in this study. The provider's years of work

experience, their monthly income (individual characteristics) and their health facility of practice (a workplace characteristic) significantly influenced $ORC_{RMC}$.

This study has provided an understanding of the state of readiness for change to RMC practice, eliminating it as a possible implementation problem for RMC practice in the study setting. We have established that $IRC_{RMC}$ and $ORC_{RMC}$ have a positive influence on each other. This study has also further confirmed the critical role of change valence and informational assessments in increasing both organisational and individual readiness for change to RMC practice. These findings have programmatic and policy implications for the designing of RMC implementation programmes. The effect of employees' perceived value for newly introduced programmes may also be evaluated on the programme intervention and implementation outcomes.

The brevity of the ORIC tool used to assess $ORC_{RMC}$ among the healthcare providers studied was also beneficial. This is in contrast to other instruments assessing organisational readiness for change with a much higher number of constructs and variables.[12 30] The ORIC tool is a standardised instrument that has been validated among health worker populations in Western countries[24 31 32] and only in South Africa,[33] with a similar population as found in our study. All categories of health providers involved in maternal care across cadres within the primary and secondary health facilities were studied. This may facilitate stakeholder engagement during the RMC implementation process and possible early adoption of the change.

The study findings however failed to establish a significant relationship between the providers' readiness for a change to RMC and their perceptions of women's rights during childbirth. RMC is premised on the fundamental human rights of women to receive dignified care.[34] It would have been expected that provider perceptions of women's rights would be positively associated with their readiness for change. The relationship was in the correct

**Table 4** Analysis of factors associated with health providers' ORC$_{RMC}$

| Covariates | Simple linear regression | | | Multiple regression | | |
|---|---|---|---|---|---|---|
| | Crude coefficient | 95% CI | P value | Adjusted coefficient | 95% CI | P value |
| Health providers' age | −0.01 | −0.02 to 0.02 | 0.916 | | | |
| Sex | | | | | | |
| Female | Ref | – | – | Ref | – | – |
| Male | −0.21 | −0.09 to 0.50 | 0.146 | 0.15 | −0.11 to 0.41 | 0.213 |
| Study Local Government Area (LGA) | | | | | | |
| Ibadan North | Ref | – | – | | | |
| Ibadan Northeast | 0.34 | −0.26 to 0.94 | 0.226 | | | |
| Ibadan Northwest | −0.22 | −0.42 to 0.02 | **0.034** | | | |
| Ibadan Southeast | −0.46 | −0.59 to 0.33 | **<0.001** | | | |
| Ibadan Southwest | −0.27 | −1.21 to 0.66 | 0.510 | | | |
| Health facility in LGA | | | | | | |
| Facility 1 | 0.43 | 0.43 to 0.43 | **<0.001** | 0.38 | 0.30 to 0.46 | **<0.001** |
| Facility 2 | Ref | – | – | – | – | – |
| Facility 3 | −0.23 | −0.23 to −0.23 | **<0.001** | 0.24 | 0.12 to 0.35 | **0.002** |
| Facility 4 | 0.65 | 0.65 to 0.65 | **<0.001** | 0.16 | −0.03 to 0.35 | **0.087** |
| Facility 5 | −0.09 | −0.09 to −0.09 | **<0.001** | −0.29 | −0.40 to −0.19 | **<0.001** |
| Facility 6 | −0.24 | −0.24 to −0.24 | **<0.001** | −0.11 | −0.16 to −0.07 | **0.001** |
| Facility 7 | 0.63 | 0.63 to 0.63 | **<0.001** | 0.56 | 0.54 to 0.57 | **<0.001** |
| Facility 8 | −0.54 | −0.54 to −0.54 | **<0.001** | −0.41 | −0.47 to −0.36 | **<0.001** |
| Facility 9 | −0.41 | −0.41 to −0.41 | **<0.001** | 0.11 | 0.02 to 0.20 | **0.024** |
| Providers' type of health facility | | | | | | |
| Primary | Ref | – | – | | | |
| Secondary | 0.09 | −0.68 to 0.50 | 0.717 | | | |
| Professional cadre | | | | | | |
| Doctor | 0.31 | −0.50 to 1.12 | 0.391 | | | |
| Nurse | −0.07 | −0.88 to 0.75 | 0.857 | | | |
| CHEW/CHO | 0.09 | −0.38 to 0.56 | 0.667 | | | |
| Health assistant/aide | Ref | – | – | | | |
| Monthly income (in US$/1000) | 0.26 | -0.05 to 0.56 | 0.083 | 0.08 | 0.02 to 0.15 | **0.021** |
| Years of professional experience/10 years | 0.05 | 0.02 to 0.3 | **0.034** | 0.08 | 0.01 to 0.2 | **0.024** |
| Years of experience in the health facility | −0.004 | −0.03 to 0.02 | 0.678 | | | |
| Awareness of the mistreatment of women | 0.02 | −0.06 to 0.09 | 0.65 | | | |
| Perceived women's rights during childbirth | 0.02 | −0.12 to 0.16 | 0.767 | | | |
| Ever heard of RMC (n=170) | | | | | | |
| Yes | 0.10 | −0.27 to 0.46 | 0.553 | | | |
| No | Ref | – | – | | | |
| Perceptions of RMC being implementable | | | | | | |
| Agreed | 0.60 | −0.02 to 1.23 | 0.056 | 0.19 | −0.08 to 0.45 | 0.148 |
| Indifferent | Ref | – | – | Ref | – | – |
| Disagreed | −0.09 | −0.76 to 0.58 | 0.765 | −0.12 | −0.60 to 0.36 | 0.57 |
| Change valence (value for RMC practice) | 0.74 | 0.47 to 1.01 | **<0.001** | 0.47 | 0.21 to 0.74 | **0.004** |

Continued

**Table 4** Continued

| Covariates | Simple linear regression | | | Multiple regression | | |
|---|---|---|---|---|---|---|
| | Crude coefficient | 95% CI | P value | Adjusted coefficient | 95% CI | P value |
| RMC Informational assessment | 0.72 | 0.40 to 1.05 | **0.001** | 0.43 | 0.22 to 0.63 | **0.002** |
| Provider perceptions on available resources | −0.002 | −0.25 to 0.25 | 0.984 | | | |
| Provider job satisfaction | 0.23 | −0.08 to 0.55 | 0.125 | 0.05 | −0.10 to 0.20 | 0.477 |
| Provider core self-evaluation | 0.15 | −0.38 to 0.68 | 0.521 | | | |
| Constant | | | | 0.06 | −1.28 to 1.41 | 0.915 |
| | | | | n=212; $R^2$=0.6016; p<0.001 | | |

Predictors with a p≤0.2 from the bivariate analysis (simple linear regression) were included in the multiple regression model. The mean variance inflation factor for the multiple regression model is=1.55. Significant p values in bold.
CHEW, community health extension worker; CHO, community health officer; $ORC_{RMC}$, organisational readiness for change to RMC; Ref, reference category; RMC, respectful maternity care.

direction but not statistically significant. The provider's low perceptions of resource availability to implement RMC did not also significantly reduce their $IRC_{RMC}$ and $ORC_{RMC}$.

This study had some limitations. It was a relatively small study and its geographical extent was limited to one metropolitan in Nigeria, which may not be representative of similar facilities and providers in other regions of Nigeria. Tertiary health facilities were not included because there was only one tertiary health facility serving populations across the five LGAs studied. Social desirability bias may have influenced some of the providers' responses positively to the availability of resources and their perception of women's rights during childbirth. To mitigate this, the data collectors stressed the academic purpose of the research to the providers when obtaining informed consent. Limited awareness of RMC, as found in this study, may affect an accurate assessment of readiness for change. We attempted to address this by educating the providers on RMC concepts before assessing their readiness for change to RMC practice.

Health providers cannot truly be ready to implement RMC if they do not support certain women's rights during childbirth. This would result in persistent mistreatment and may prevent a positive change to RMC practice. The most common forms of mistreatment to women during childbirth in the study health facilities were being denied birth companions, not being allowed to decide on a birth position and being denied mobility in labour. All three forms of mistreatment were also reported by Tanzanian women in a qualitative study of the perspectives of mothers and fathers on mistreatment during childbirth.[35] Several other studies have reported these forms of mistreatment experienced by women during childbirth.[36–39] According to the WHO,[40 41] having a birth companion during labour provides emotional support, reduces labour pain and strengthens the woman's capability to deliver. The WHO has also recommended that women are supported to deliver in their preferred birth position because alternative birth positions, such as standing to deliver, are safe and may result in shorter labour from better fetal alignment.[38 40] It has also been reported that mobility during the first stage of labour is safe.[38] Denying women autonomy, or not respecting women's choices during childbirth without a justifiable medical reason, constitutes mistreatment that negatively affects their overall childbirth experience.[42]

The health providers perceived that women should always have the right to full information about their care and to receive their care in privacy. Unfortunately, many may not practise it for several reasons, including unconscious behaviour, an abusive work culture and perceived excessive workload among others.[43] About 33% of maternity care providers in Western Kenya attested that they do not often give explanations before conducting procedures on women during childbirth, and 73% do not wait to obtain consent before conducting these examinations.[43] This is similar to the inconsistent support for women's right to autonomy found among Australian midwives and doctors.[44] They confirmed their support for women's autonomy, but over-ride women's decisions sometimes on safety reasons, claiming full accountability for every pregnancy outcome. Women should be included when safety decisions are being made during childbirth. When this is not done, women may conclude it is an abuse of their rights. Tanzanian women related their abusive maternity care experiences as a deviation from their basic human rights.[45] Hence, advocating for women's rights among health providers should be a key component of RMC-promoting interventions.

Nonetheless, the health providers scored high in their perceived $IRC_{RMC}$ and $ORC_{RMC}$. Few studies had reported the overall ORIC in health programmes as mean scores using the ORIC tool. Many either report the mean change commitment and change efficacy as individual scores,[21] or as total scores.[31] The $ORC_{RMC}$ score in our study was higher than the average of the change commitment and change efficacy scores found when the nurse-reported

organisational readiness for change for policy change in acute care hospitals in Switzerland was assessed.[21] There was no comparable study of individual readiness for change using the same instrument applied in the health industry. A scoping review to explore the nature and extent of literature published on individual readiness for change in the health sector yielded no study found in health.[46]

$IRC_{RMC}$ and $ORC_{RMC}$ in our study were significantly positively correlated. Thus, a positive increase in $IRC_{RMC}$ by strengthening its facilitating factors should also reflect in increased $ORC_{RMC}$. This is similar to the postulations by Weiner in his theory where he stated that 'Organisational readiness is likely to be highest when organisational members not only want to implement an organisational change but also feel confident that they can do so' (page 3).[11]

Weiner theorised that organisational readiness was most strongly influenced by change valence and informational assessments.[11] The health providers' change valence positively influenced both their $IRC_{RMC}$ and $ORC_{RMC}$ significantly in our study. Change valence also positively and significantly influenced organisational readiness for change among employees of a private hospital changing to a tertiary hospital.[47] It also strongly correlated with individual readiness for change in the automobile industry.[48] There have been limited assessments of individual readiness for change in health-related industries.

Informational assessment is the perceived adequacy of the available resources such as the equipment, expertise, skills and time needed to implement the change. Informational assessments also significantly influenced both $IRC_{RMC}$ and $ORC_{RMC}$ in this analysis. Informational assessment of their perceived resource adequacy was found to be positively and significantly correlated with their perceived resource availability in this study. This suggests that if providers' perception of resource availability is high, they would be ready for a change to RMC practice. However, the providers had a low perception of the availability of recommended resources for RMC implementation in our study setting. This may have explained their fairly low perceived resource adequacy.

Thus, additional resource requirements are critical drivers of RMC implementation.[11] For example, only 9% of the health providers agreed that facilities to support birth companions were available. This would include a private space achievable with the use of curtains. In an observational study of childbirths across four countries, Nigeria had the lowest proportion of women (6.9%) in which curtains were used to ensure privacy.[49] This is a challenge that may prevent Nigerian women from receiving RMC as there is limited funding to the Nigerian health system to provide these essential RMC resources. There is a need to identify cost-effective strategies to address these system challenges.

$ORC_{RMC}$ was found to be significantly higher among health providers with longer years of work experience. They are a population to target in RMC-promoting interventions. The nurses' years of work experience also positively influenced their change commitment, one of the measures of organisational readiness, in Switzerland's acute care hospitals.[21] The providers' workplace setting, as indicated by their health facility of practice, significantly influenced their perceived $ORC_{RMC.}$ This was significantly positive for most of the primary healthcare facilities across the LGAs and was significantly negative for two of the secondary health facilities studied. Interestingly, both the primary and secondary health facilities in the Ibadan Northwest LGA were significantly associated with a decreased $ORC_{RMC}$. According to the literature, the workplace contextual fit is critical to providers' readiness for change to RMC as it informs the adaptability of the local context to the globally defined RMC practice, the quality of the implementation and whether expected RMC implementation outcomes will be achieved.[50–52] There is the need to qualitatively explore which contextual factors within the health facilities are the most critical barriers to a successful implementation of RMC practice during childbirth.

## CONCLUSIONS

The three most common forms of mistreatment during childbirth noted by health providers corresponded with the low recognition of these as rights that women should always receive. Our study confirmed the relevance of the organisational and individual readiness for change constructs to the RMC literature and should prompt more studies on this topic. It is noteworthy that the health providers in our study perceived themselves and their organisations to be ready for a change to RMC practice. It would be important to verify in future research if readiness for change significantly facilitated the implementation of RMC interventions. The main influencing factors of both $IRC_{RMC}$ and $ORC_{RMC}$ scores in our analysis were a high valuation of the change (change valence) and the perceived adequacy of resources necessary to implement the change. Longer serving providers may be a readier population to target during RMC implementation, as champions to lead a change to RMC practice. Workplace contexts could significantly influence $ORC_{RMC}$ and should be explored before the implementation of RMC interventions.

**Acknowledgements** We acknowledge the contributions of Dr Tunde Adedokun, Dr Joshua Akinyemi and Professor Folusho Owotade for their inestimable advice and support with the data analysis. We also recognise and appreciate all the health providers who completed the survey despite their hectic schedules. SM acknowledges the funding support by the South African Medical Research Council's Mid-Career Scientist Award.

**Contributors** OTE conceptualised the study, designed the study, acquired and adapted the tools to the study and obtained the ethical approvals. She was the principal investigator who conducted the data collection and supervised the research assistants. She analysed the data and wrote the first draft and final manuscripts for publication. OTE also serves as the guarantor for the research. SM contributed to the design of the study and the finalisation of the tools. She also significantly contributed to the revision of the draft manuscript and approved the

final manuscript for publication. DB contributed significantly to the design of the study, the finalisation of the tools and the data analysis. He significantly revised and contributed significant intellectual content to the draft manuscript and approved the final version of the manuscript for publication.

**Funding** This research was supported by the Consortium for Advanced Research Training in Africa (CARTA). CARTA is jointly led by the African Population and Health Research Centre and the University of the Witwatersrand and funded by the Carnegie Corporation of New York (grant no: G-19-57145), Sida (grant no: 54100113), Uppsala Monitoring Centre, Norwegian Agency for Development Cooperation (Norad), the Wellcome Trust (reference no. 107768/Z/15/Z) and the UK Foreign, Commonwealth & Development Office (no grant number), with support from the Developing Excellence in Leadership, Training and Science in Africa (DELTAS Africa) programme (no grant number).

**Disclaimer** The funders only provided the funding for the research. They played no role in the conceptualisation, design and conduct of the research nor the data analysis, interpretation and development of the manuscript for publication.

**Competing interests** None declared.

**Patient and public involvement** Patients and/or the public were involved in the design, or conduct, or reporting, or dissemination plans of this research. Refer to the Methods section for further details.

**Patient consent for publication** Not required.

**Ethics approval** This study involves human participants and ethical approvals were obtained from the Human Research Ethics Committees (HRECs) of the University of the Witwatersrand, Johannesburg (clearance number M190658), and the Oyo State Ministry of Health (ref. number AD/13/479/1386). Participants gave informed consent to participate in the study before taking part.

**Provenance and peer review** Not commissioned; externally peer reviewed.

**Data availability statement** Data are available in a public, open access repository. The dataset generated and analysed in the current study is available from the Figshare database. The doi is 10.6084/m9.figshare.19757329.

**ORCID iD**
Oluwaseun Taiwo Esan http://orcid.org/0000-0002-2908-6034

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
