## [Reviewer comments · BMJ Open]

ARTICLE DETAILS

TITLE (PROVISIONAL)	Organisational and individual readiness for change to respectful maternity care practice and associated factors in Ibadan, Nigeria: a cross-sectional survey
AUTHORS	Esan, Oluwaseun; Maswime, Salome; Blaauw, Duane

VERSION 1 – REVIEW

REVIEWER	Mdoe, Paschal Haydom Lutheran Hospital, Obstetrics and Gynecology
REVIEW RETURNED	05-Jul-2022

GENERAL COMMENTS	Thank you for allowing me to review this manuscript. You have touched very important topic. You have done good work You need to review the sample size in the Abstract you have 212, for sample calculation, you have 210 and in the methods text, you have 244. I think you need to clear this Some of the information in the discussion would fit the background. You may need to re check
--

REVIEWER	Okonofua , F University of Benin, Obstetrics and Gynaecology
REVIEW RETURNED	06-Aug-2022

GENERAL COMMENTS	Article: Organizational and individual readiness for change to respectful maternity care practice and associated factors in Ibadan, Nigeria: A cross-sectional study Review Originality The paper reports the results of a study that assessed health providers organizational and individual readiness for change to respectful maternity care in Ibadan, Southwest Nigeria. The study is novel given that not much has been published previously on preparedness for respectful maternity care especially within the context of African countries. With an increasing number of publications relating to the high prevalence of disrespectful maternity care in Africa, this study's focus on the preparedness of health providers to deal with the problem is commendable. However, the failure of the paper to go further to elucidate how individual and organizational preparedness can be taken to action, limits the policy applicability of the paper. Scientific Merit I have additional comments as follows: • The authors need to explain the rationale for the choice of Ibadan as the location for the study. Perhaps, this can be linked to the
---

	previous study they conducted in the area, for which they need to provide some idea of the quantum and nature of disrespectful maternity care in the area that justify the study.  • The background to the paper needs to be better elucidated. I believe more international, regional, and national data on disrespectful maternity care ought to be provided to provide substantive justification for the study. By contrast, the introduction focused to a large measure on describing the conceptual framework, without providing sufficient evidence on previous studies undertaken on the subject matter. I believe that the aspects dealing with conceptual framework should be removed from the introduction; the introduction should be expanded to justify the study; while the conceptual framework should be provided in a separate sub-heading after the introduction. I also suggest that some form of a theory of change should be embedded to explain how this potentially novel study can lead to change in promoting respectful maternity care in the hospitals. • The method section needs some revision. As an example, the sample size of 210 was determined by using the one-sample test mean in Stata. This needs to be explained further by identifying the variables and forecasts that enabled the sample size determination • The paragraph of the methods beginning with “For the predictors, Weiner”---- and ending with “perception of women’s rights” (beginning from line 150) should be removed from the methods section as it does not explain the methods used for the study. If possible, the section could be moved to the Discussion part of the paper. • My serious concern with this paper is that the authors used the composite indicators of “commitment to change” and “change efficacy” to measure organizational readiness for change. I seriously doubt if these two components can explain organizational readiness in full. I believe organizational readiness would be better explored by assessing the organizational governing structure, and the extent to it is willing and able to deploy resources to manage the change process. While individuals may be committed to change, they will be unable to drive the change if the organization and its managers do not commit themselves to management of the change process • I believe that this study would have been more informative if it included a qualitative component and interviews with the heads and managers of the institutions (mixed study design) to enable the substantive assessment and determination of organizational readiness. • The discussion part of the paper needs considerable improvement to explain the method used as well as the results. • Also, the policy implications and public health relevance of the study should be better presented. • There are several typos and grammatical errors in various parts of the paper that need to be corrected. Recommendation Accept after major revision Remarks to the editor The data analysis section is rather detailed. As I am not an expert in data analysis, I would advise that this section should be reviewed by a statistician
--	--

VERSION 1 – AUTHOR RESPONSE

Table: Response to the editor and reviewers' comments

S/no	Reviewers' comments	Response	Line & Page No
	Editorial comments		
1	Thank you for including a STROBE checklist with your submission. In addition, along with your revised manuscript, please also include a copy of the CROSS checklist for the reporting of survey studies indicating the page/line numbers of your manuscript where the relevant information can be found (https://www.equator-network.org/reporting-guidelines/a-consensus-based-checklist-for-reporting-of-survey-studies-cross/), updating the manuscript as needed to ensure all reporting requirements are met.	This has been done and the CROSS checklist has been included in addition to the STROBE's checklist	Line 492, Page 23 & Additional files
2	Please update the article title and the abstract 'Design' section to refer to the study as a survey, since this is a more specific description of the study design and is used elsewhere in the manuscript.	A survey design has been added to the title. As well as the method section of the abstract	Line 2, Page 1 & Line 32 page 2
3	Please update the abstract Results section to include numerical data, including measures of statistical significance, for all reported findings. Please also note that p values should generally not be reported in isolation, as readers need to be able to appreciate the magnitude of the findings, not just the statistical significance – please amend accordingly in the abstract (and elsewhere as needed).	This has been addressed. The numerical values such as the p-values and 95% confidence intervals (CI) have all been included for all the results provided.	Lines 46, 49-53 (pages 2 & 3)
4	*Please revise the 'Strengths and limitations of this study' section of your manuscript (after the abstract). This section should contain up to five short bullet points, no longer than one sentence each, that relate specifically to the methods. The novelty, aims, results or expected impact of the study should not be summarised here. Points referring to this being 'the first study' are not appropriate, as these concern the novelty of the study, rather than a methodological strength.	This has been done. The bulleted points now speak solely to the methodology of conducting the research. The last two bullets speaking to the limitations were retained	Lines 60-70, Page 3
5	Please include a copy of the survey questionnaire instrument (in English) as a supplemental file, cited in the main text	This has been added and referred to in the text	Line 163, page 7 and

	Methods. This should exclude any copyrighted materials, which should instead be cited only.		Additional file 2
6	Please revise the 'Availability of data and materials' statement. This includes two sentences that at present seem redundant. If all the data are on the figshare website, what is the additional material that will be shared on reasonable request? Please revise to clarify which data are on the figshare website and which additional data will be shared by the authors on reasonable request.	This has been done. The last two sentences in this section have been deleted.	493-495, page 23
7	*Please complete a thorough proofread of the text and correct any spelling and English-language grammar and phrasing errors that you identify.	This has been done	
	Reviewer 1		
8	You need to review the sample size in the Abstract you have 212, for sample calculation, you have 210 and in the methods text, you have 244. I think you need to clear this	The 244 is the total study population, the denominator. The total number of health workers in the sampled facilities. The calculated sample size is 210 (86% of the study population) While 212 were interviewed (101%) of the sample size. These have also been clarified in the text.	Lines 139, page 6 Line 141-142-153, page 6 Lines 219-220, page 9
9	Some of the information in the discussion would fit the background. You may need to re-check	These have been re-checked All the information provided in the discussion section was to discuss the findings.	
10	Reviewer 2		
11	The authors need to explain the rationale for the choice of Ibadan as the location for the study. Perhaps, this can be linked to the previous study they conducted in the area, for which they need to provide some idea of	The rationale for the choice of Ibadan as the study location has been provided in the method section	Lines 213-215, page 9

	the quantum and nature of disrespectful maternity care in the area that justifies the study		
12	The background to the paper needs to be better elucidated. I believe more international, regional, and national data on disrespectful maternity care ought to be provided to provide substantive justification for the study. By contrast, the introduction focused to a large measure on describing the conceptual framework, without providing sufficient evidence on previous studies undertaken on the subject matter. I believe that the aspects dealing with conceptual framework should be removed from the introduction; the introduction should be expanded to justify the study; while the conceptual framework should be provided in a separate sub-heading after the introduction. I also suggest that some form of a theory of change should be embedded to explain how this potentially novel study can lead to change in promoting respectful maternity care in the hospitals.	An introductory paragraph has been added to the background section addressing national data on the mistreatment of women. We described the concept of readiness for change. This was to give a background to understanding the study. And it was intertwined with the justification of the study. So we could not cut out a separate section for the conceptual framework. We have added a paragraph describing the proposed Theory of Change for the research.	Lines 74-83, page 4 Lines 92 - 103, pages 4-5. Lines 104-113, page 5 Lines 114-120, page 5
13	The method section needs some revision. As an example, the sample size of 210 was determined by using the one-sample test mean in Stata. This needs to be explained further by identifying the variables and forecasts that enabled the sample size determination	More detail has been added to address this comment.	Lines 142-145, page 6
14	The paragraph of the methods beginning with “For the predictors, Weiner”---- and ending with “perception of women’s rights” (beginning from line 150) should be removed from the methods section as it does not explain the methods used for the study. If possible, the section could be moved to the Discussion part of the paper.	We tried moving the sections as advised, but they could not fit the discussion nor the background sections. We have rewritten the section to properly read and describe the methodology that it is. The section described the variables	

		(predictors) measured. The section described both the previously defined predictors and the newly proposed or added ones measured in this study.	Lines 176-182, page 8
15	My serious concern with this paper is that the authors used the composite indicators of “commitment to change” and “change efficacy” to measure organizational readiness for change. I seriously doubt if these two components can explain organizational readiness in full. I believe organizational readiness would be better explored by assessing the organizational governing structure, and the extent to it is willing and able to deploy resources to manage the change process. While individuals may be committed to change, they will be unable to drive the change if the organization and its managers do not commit themselves to management of the change process	Thank you for this comment We explored the contextual factors and barriers that may limit readiness for change to RMC practice qualitatively by conducting However, the findings are to be published in another article. This is because of the limited word count and because there are lots of issues to be addressed from the findings. While the readiness for change assessed and presented here was a lot to address already. So, we have added the need to identify and address the contextual factors and barriers before the expected change can indeed be actualised. We also recommended the need for future research on these.	Lines 118-120, page 5 Lines 462-463, page 22
16	The discussion part of the paper needs considerable improvement to explain the method used as well as the results.	This has been done. We have also introduced a paragraph that discussed the methods used.	355-363, pages 17-18.
17	Also, the policy implications and public health relevance of the study should be better presented.	We have elucidated some of the programmatic and policy implications of some of the findings the more.	Lines 352-354, page 17. &

			Lines 474-477, page 22
18	There are several typos and grammatical errors in various parts of the paper that need to be corrected	These have been checked and corrected all through the manuscript	

VERSION 2 – REVIEW

REVIEWER	Okonofua , F University of Benin, Obstetrics and Gynaecology
REVIEW RETURNED	18-Oct-2022
GENERAL COMMENTS	Congratulations on your successful and elegant revision of the paper